

# Effect of vascular endothelial growth factor 165 on dopamine level in the retinas of guinea pigs with form-deprivation myopia

Ruiting Sun[1,*], Qingsheng Peng[1,*], Fengyi Zhang[1], Honglian Gao[2], Tong Li[1], Lei Wang[1] and Lei Zhang[1]

[1] Department of Ophthalmology, Binzhou Medical University Hospital, Binzhou, Shandong Province, China
[2] Medical Research Center, Binzhou Medical University Hospital, Binzhou, Shandong Province, China
* These authors contributed equally to this work.

Corresponding author
Lei Zhang, zhangleisd@bzmc.edu.cn

## ABSTRACT

**Background:** Myopia is the most common refractive error because excessive increase in the axial length of a myopic eye leads to the thinning of the posterior scleral pole and can cause serious complications resulting in blindness. Thus, myopia has become a great concern worldwide. Dopamine (DA) plays a role in the development of myopia. Moreover, in Parkinson's disease, it has been proved that vascular endothelial growth factor 165 (VEGF165) can promote the survival and recovery of DA neurons, resulting in increased DA secretion in the striatum, thereby treating neuropathy. Therefore, we speculate that VEGF165 can also promote the release of DA in the retina to inhibit the occurrence and development of myopia. We aimed to investigate the effect of VEGF165 on DA levels in the retinas of guinea pigs with form-deprivation myopia (FDM) and the effects of DA on myopia prevention and control.

**Methods:** Healthy 3-week-old pigmented guinea pigs were randomly divided into blank, FDM, phosphate buffer saline (PBS), 1, 5, and 10 ng groups. The FDM model was established by covering the right eye continuously with a translucent latex balloon pullover for 14 days. The pigs in the PBS, 1, 5, and 10 ng groups were injected with PBS buffer and 1, 5, and 10 ng of VEGF165 recombinant human protein, respectively, in the vitreous of the right eye before masking. The refractive error and axial length were measured before and after modeling. All retinas were used for biomolecular analyses after 14 days.

**Results:** We found that the intravitreal injection of VEGF165 elevated DA levels in the retina and was effective in slowing the progression of myopia, and 1 ng of VEGF165 was the most effective. Moreover, the number of vascular endothelial cell nuclei in the 1 ng group was lower than that in the other VEGF165 groups.

**Conclusions:** Our data suggest that VEGF165 has a promoting effect on DA in the retinas of guinea pigs with FDM, potentially controlling the development of myopia.

## INTRODUCTION

Myopia is the most common form of refractive error. Myopia is a condition in which eye adjustment is relaxed and parallel rays enter the eye, the rays are focused before they reach the retina, and a clear image cannot form on the retina because of excessive corneal curvature or excessive length of the eyeball (*Flitcroft et al., 2019*). Myopia can be divided into refractive myopia and axial myopia, in which the length of the eye axis is critical (*Young, Metlapally & Shay, 2007*). Excessive increase in the axial length (AL) of myopic eyes lead to the remodeling of scleral tissues and the accelerated thinning of the posterior scleral pole (*Curtin, Iwamoto & Renaldo, 1979*), causing serious complications, such as retinal tearing, retinal detachment, and myopic macular degeneration, which can lead to blindness (*Verhoeven et al., 2015*). The prevalence of myopia is gradually increasing globally and expected to rise to more than half of the global population by 2050 (*Holden et al., 2016*). Therefore, identifying the cause of myopia and controlling its progression are important.

The animal models of myopia are important means to study the pathogenesis of myopia. Many neurotransmitters in the retina are involved in the occurrence and development of myopia, such as dopamine (DA), choline, nitric oxide, and γ-aminobutyric acid (*Wang et al., 2021*). DA is an essential neurotransmitter in the retina and mediates a variety of functions, including development, visual signaling, and refractive development (*Witkovsky, 2004*). Retinal DA is synthesized in and released by the subtypes of amacrine and interplexiform cells (*Dowling & Ehinger, 1978a*, *1978b*; *Frederick et al., 1982*). Data from several experiments on different species suggest that DA acts as a "stop" signal in refractive eye growth (*Feldkaemper & Schaeffel, 2013*; *Nebbioso et al., 2014*). In the form-deprivation myopia (FDM) model experiments of different species, primates, chickens, and guinea pigs showed decreases in DA levels in their retinas during FDM (*Dong et al., 2011*; *Iuvone et al., 1989*; *McBrien & Gentle, 2001*; *Papastergiou et al., 1998*; *Stone et al., 1989*). Moreover, increase in intraocular DA level can prevent the formation of FDM in guinea pigs (*Mao et al., 2010*). Therefore, DA in the retina plays a key role in the development of myopia.

VEGF, as a member of a large family of angiogenic factors, is a peptide growth factor closely related to neovascularization; VEGF not only promotes angiogenesis and enhances vascular permeability (*Wang, Dentler & Borchardt, 2001*; *Zhang et al., 2000*), but also exerts neurotrophic and neuroprotective effects (*Ortuzar et al., 2011*, *2010*; *Storkebaum, Lambrechts & Carmeliet, 2004*). VEGF has seven subtypes (*Ferrara & Adamis, 2016*), and VEGF-A (referred to as VEGF in this article) is the main form. VEGF is spliced by selective exons to generate a variety of isomers, including VEGF121, VEGF165, VEGF189, and VEGF206, of which VEGF165 is the most physiologically relevant isomer (*Darrington et al., 2012*). VEGF has neuroprotective effects on DA neurons *in vitro* and *in vivo* (*Yasuhara et al., 2004*), and this neuroprotective effect is attributed to its dual effect on nerve cells and blood vessels. *In vitro*, VEGF can directly promote the survival of DA neurons, and *in vivo*, VEGF can indirectly protect DA neurons by improving microcirculation through angiogenesis and nutritional factor release and can increase

vascular permeability (*Yasuhara, Shingo & Date, 2004*). However, the effects of VEGF are dose independent: the greatest benefit is obtained at low doses of VEGF, and high doses cause excessive angiogenesis and tissue edema (*Yasuhara et al., 2005b*). The current experiment will explore the intravitreal injection of different doses of VEGF165 in FDM guinea pigs and study its effect on retinal DA level and angiogenesis. The aim is to obtain insights and possible drug treatment methods for controlling the occurrence and development of myopia.

## MATERIALS AND METHODS

### Experimental animals

A total of 90 healthy 3-week-old pigmented guinea pigs (male or female, 150–200 g) were purchased from Jinan Jinfeng Laboratory Animal Corporation randomly divided into six groups of 15 pigs in each group. The light environment of the animal was approximately a 12 h day/12 h night cycle, and the room temperature was maintained at 22–25 °C at 50–70% humidity. All animals had free access to food and water. The study was approved by the ethics committee of the Affiliated Hospital of Binzhou Medical University (approval number: 20220128-78), and the feeding and use of experimental animals strictly followed the relevant regulations of the animal management committee of Binzhou Medical University. All guinea pigs were euthanized with intraperitoneal injection of three times anesthetic dose (1% Pentobarbital sodium 120 mg/kg). During the experimental period in which disease signs (including weakness in the extremities, lethargy, lack of food intake, or unexpected infection) were detected in FDM models, 1% Pentobarbital sodium 120 mg/kg was given to guinea pigs for deep anesthesia, and then guinea pigs were euthanized.

### Induction of form-deprivation myopia and injections

A white translucent nontoxic latex balloon was placed on the head of each guinea pig in a manner that it only covers the right eye of the guinea pig and fully exposes the left eye, mouth, nose, and ears. The balloons were washed once a day to keep the hood clean. An FDM guinea pig myopia model was established by continuous masking for 14 d.

Recombinant human protein VEGF165 (100-20; PEPROTECH, Cranbury, NJ, USA) was dissolved in sterile water for long-term storage at a concentration of 1 ng/μL in a −80 °C freezer. Stock solutions were diluted with phosphate buffer saline (PBS) solution immediately before use. Before FDM treatment, PBS solution (2 μL) and recombinant human protein VEGF165 (1, 5, and 10 ng VEGF165 were dissolved in 2 μL PBS solution) were injected into the vitreous of the right eye of each guinea pig. Tropicamide phenylephrine eye drops were used 20 min before surgery to open the pupils. After the intraperitoneal injection of anesthetics (1% Pentobarbital sodium 40 mg/kg), the eye areas and ocular surfaces of the guinea pigs were fully disinfected. The eyeball was fixed with microscopic toothed forceps under the microscope. The microsyringe was injected vertically downward 1 mm behind the temporal limbus, and the needle was tilted 20° toward the eyeball wall to avoid the lens. The microsyringe was further advanced about 3 mm, and the drug was injected into the vitreous. The needle hole was gently clamped

with forceps for 30 s after injection. Ofloxacin eye ointment and sterile gauze dressing were applied after the operation. All experiments were conducted by experienced investigators.

## Ocular measurements

Ocular refraction and AL were recorded in both eyes of each animal before and 2 weeks after treatment. All measurements were carried out under minimal lighting after facemask removal. Before a measurement, tropicamide phenylephrine eye drops were applied to the right eye, and refraction was measured by a streak retinoscope (YZ24; 66 Vision-Tech, Suzhou, China) after the pupil was fully dilated. The equivalent spherical lens (spherical degree + 1/2 cylindrical lens degree) was used for data analysis, and the average value of five measurements was used as the refractive value.

Propmecaine hydrochloride eye drops were applied to the right eye before the length of the eye axis was measured and after anesthesia was applied on the surface of the cornea. Ultrasound (AVISO; Quantel Medical, Cournon-d'Auvergne, France) was used to measure the length of the eye axis when the probe was aimed at the center of the cornea and perpendicular to the plane of the cornea. The length of the eye axis of the right eye was continuously measured five times in manual mode, and the average value was calculated to an accuracy of 0.01 mm.

## HPLC analysis

All guinea pigs were sacrificed *via* euthanasia. The removed eyeball was placed on ice, and retinas were harvested within 30 s under a dissecting microscope, immediately frozen in liquid nitrogen, and stored at $-80\,°C$ until all the tissue samples were collected. Each milligram of frozen sample was added to 20 µL of freshly prepared homogenate (0.1 M $H_3ClO_4$, 0.1 mM EDTA $Na_2$, and internal standard DHBA) and frozen at $-40\,°C$. After low-temperature centrifugation (20,000 rpm, 30 min, $4\,°C$), the supernatant was collected and tested on the machine. An HPLC system (Thermo UniMate 3000 Pump, flow rate 0.2 ml/min, ESA Coulochem III. Electrochemical Detector, 5041 Cell voltage 350 mV, Guard Cell voltage 360 mV, Rang 100 nA, Filter 10S; Thermo Fisher Scientific, Waltham, MA, USA) was used to measure the levels of DA and DOPAC. The column was Thermo Acclaim rapid separation liquid chromatography (2.1 mm × 100 mm; C18, 2.2 µm; column temperature, $40\,°C$). The mobile phase contained $NaH_2PO_4$ (90 mM), citric acid (50 mM), OSA (1.7 mM), EDTA (50 µM), and acetonitrile (4.5%). Data were acquired and analyzed using a Chromeleon 6.9 chromatography workstation, and target concentrations were calculated using internal standards.

## Immunofluorescence analysis

The right eyeball was removed on ice, and the excess connective tissue and extraocular muscles were removed under the microscope to preserve the optic nerve and maintained the integrity of the eyeball. For immunofluorescence, the eyes were fixed in Bouin's fixative solution (PH0976; Scientific Phygene, Fuzhou, China) for 24 h and then transferred to a 20% sucrose solution. After 12 h, 30% sucrose solution was added. The solution was left to stand for 12 h at $4\,°C$. The anterior segments of the eyes were removed, and the eyecups

were embedded in optimum cutting temperature media (Order Number 4583; SAKURA, Torrance, CA, USA) and then frozen at −80 °C for long-term storage. The frozen eyecups were sectioned vertically on a freezing microtome (CM1950; Leica, Wetzlar, Germany) at a thickness of 8 μm. The tissue sections were adsorbed on viscous slides, treated with acetone at 4 °C, washed three times with PBS solution, and rinsed three times with 0.3% Trition X-100. Retinal sections were blocked with 5% albumin bovine (Biosharp, Anhuim, China) in a 37 °C incubator for 30 min. The sections were then incubated with primary antibodies (TH, 1:100; Cell Signaling Technology, Danvers, MA, USA) overnight at 4 °C, followed by appropriate fluorophore-conjugated secondary antibodies (1:500; Abcam, Cambridge, UK) for 1 h at 37 °C in the protected state. After thorough washes, each sample was gently mounted with a cover glass, observed the number of TH-positive cells under an upright fluorescence microscope (BX51+DP72; Olympus, Tokyo, Japan), and photographed. The number of TH-positive cells were counted using a fluorescence microscope under a ×20 objective. To assess the number of TH-positive cells under immunofluorescence microscopy, positive cells of the photomicrographs obtained from retinal sections of different experimental groups were measured. Data obtained in six cross sections were averaged for one sample, and six samples were collected for each differently manipulated group.

## Western blot analysis

All guinea pig eyeballs were removed, and the retinas were harvested and weighed. RIPA lysis buffer (Biosharp, Anhui, China) and protease inhibitor cocktail (MedChemExpress, Monmouth Junction, NJ, USA) was added to the tissues, and an ultrasonic crusher was used to further fragment and homogenize the tissues. The samples were centrifuged at 12,000 r/min for 30 min in a 4 °C centrifuge (TGL-20R; Shan Dong BaiO Medical technology CO., LTD., Shandong, China), and the supernatant was collected.

The concentration was quantified using a BCA protein assay kit (Biosharp, Anhui, China). All the samples were boiled with 5× loading buffer at 95 °C for 10 min. From each group of retinal tissues, 20 μg of protein homogenate was extracted and added to 10% sodium dodecyl sulfate polyacrylamide gel for electrophoresis and transferred to 0.45 mm-thick polyvinylidene fluoride membrane. Nonspecific membrane binding was blocked with 5% BSA for 2 h at room temperature. The membranes were incubated with the primary antibodies anti-TH (1:1,000; Cell Signaling Technology, Danvers, MA, USA), anti-CD31 (1:1,000; Abcam, Cambridge, UK), and GAPDH (1:10,000) overnight at 4 °C. After the membranes were washed with Tris-buffered saline containing 0.1% Tween-20, they were incubated against the secondary antibody (goat anti-rabbit 1:10,000) for 2 h at room temperature. TH and CD31Protein bands were visualized using a ChemiScope Capture Image acquisition software (ChemiScope 6200 Touch; Clinx Science Instruments Co., Ltd., Shanghai, China), and ImageJ was used to analyze the optical density. Each experiment was repeated three times.

### Hematoxylin and eosin (HE) staining

Frozen retinal tissues were cut into 8 μm sections. The retinal sections were fixed by acetone at 4 °C and then washed with 1% PBS for 5 min three times. All the samples were stained with hematoxylin and eosin (HE), dehydrated with ethanol and xylene, and sealed with neutral glue. The number of vascular endothelial nuclei distributed within the retinal boundary membrane were counted using a microscope under ×20 objectives. Six samples were randomly selected from each differently manipulated group and statistically analyzed after calculating the mean number of vascular endothelial cell nuclei.

### Statistical analysis

The statistical software SPSS 25.0 was used for statistical analysis. All data were confirmed by Shapiro–Wilk test to be normally distributed and expressed as mean ± SD.
The between-group data were homogeneous with variance according to the Levene's test. One-way ANOVA was used for the overall comparison of the data in each group. LSD-t test was performed to compare the groups. $P < 0.05$ indicated statistical significance.

## RESULTS

### Refraction error and ocular parameters

Before the experiment, all guinea pigs in each group showed hyperopia in both eyes. No significant difference in refraction or AL was found between the right and left eyes of each animal. After 2 weeks of FDM treatment, the FDM eyes showed significant ocular AL elongation and myopic shift (refraction error, blank eye *vs*. FDM eye, +2.417D *vs*. −3.083D, AL, blank eye *vs*. FDM eye, 7.710 mm *vs*. 8.120 mm, $P < 0.05$) (Figs. 1A and 1B).

To evaluate the efficacy of the three concentrations of recombinant human VEGF165 in attenuating progressing myopia, PBS buffer and 1, 5, and 10 ng of recombinant human protein VEGF165 were injected intravitreal into the eyes of the FDM guinea pigs. The groups with intravitreal injection of recombinant human VEGF165 had significantly shorter AL than the groups with intravitreal injection of PBS buffer and FDM (Fig. 1D). Moreover the refractive error of the three VEGF groups were significantly hyperopic compared with that of the PBS buffer and FDM groups (Fig. 1C). Moreover, 1 ng of recombinant human VEGF165 was the most effective in attenuating myopia progression in terms of AL elongation and refractive error.

### Expression of DA and DOPAC levels in the retina

We first explored whether the retinal DA levels in the eyes form deprived for 2 weeks were reduced in guinea pigs. The levels of DA and DOPAC in the retina of the guinea pigs in the FDM group were significantly lower than those in the normal eyes according to the HPLC assay, and the differences were statistically significant ($P < 0.05$). Three different concentrations of VEGF165 were injected into the vitreous of the guinea pigs. The most significant increases in the levels of DA and DOPAC were observed in the retinas of the guinea pigs in the 1 ng group, followed by those in the 5 ng groups, and those of the 10 ng groups had the least significant increases (Figs. 2A and 2B). No significant difference in

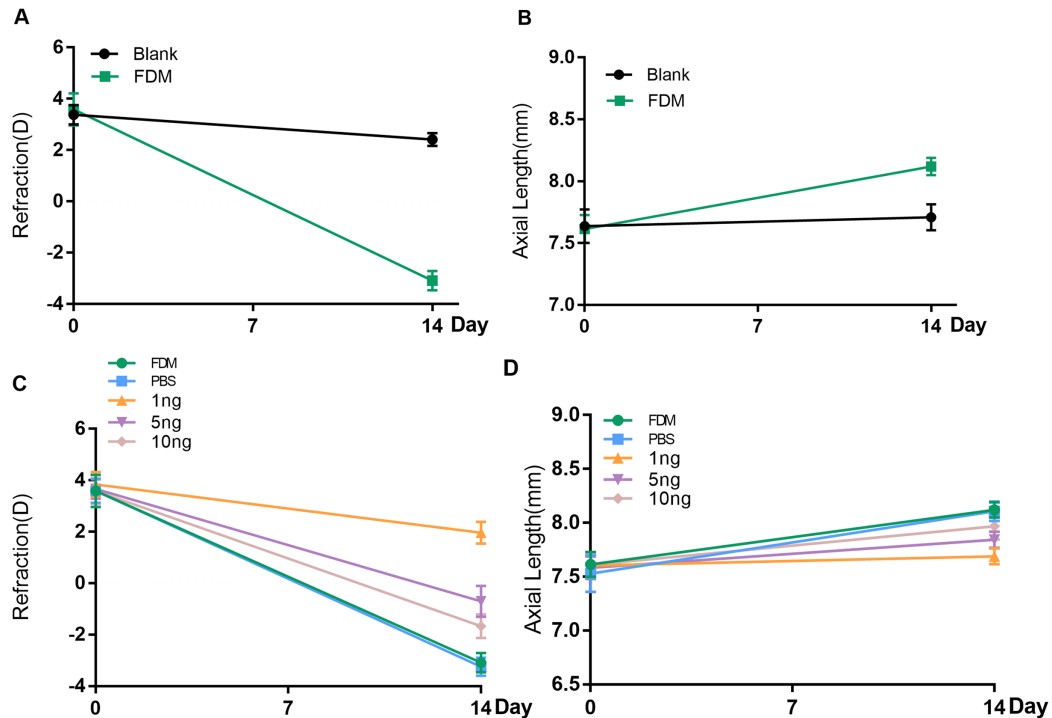

**Figure 1 Changes in refraction or AL in each group after 14 days.** (A and B) Represent changes in refraction and axial length (AL) of the right eye of guinea pigs without any intervention and the right eye of the FDM model after 14 days ($n = 6$ for Blank group and 6 for FDM group). (C and D) Showed that the three groups with different doses of VEGF165 had inhibitory effects on the refraction and AL of FDM eyes after 14 days, among which the 1 ng group had the most obvious inhibitory effect, and there was no significant change in the PBS group compared with the FDM group. Graphs were presented as mean ± standard deviation. The data were analyzed using LSD-t test.

DOPAC/DA (DA metabolic rate) in the retina was found among all groups ($P > 0.05$) (Fig. 2C).

## Number of TH$^+$ cells in immunofluorescence

Given that TH is a rate-limiting enzyme for DA synthesis, we counted the number of TH$^+$ cells in the retina to reflect the amount of DA. The immunofluorescence results showed that the expression of TH$^+$ cells in the retina was green punctate fluorescence. TH$^+$ cells were located in the plane of the inner nuclear layer, where the somata of DA amacrine cells are located (Fig. 3A). The results of pairwise comparison showed that the number of TH$^+$ cells in the FDM group was significantly lower than that in the blank group. The number of TH$^+$ cells in the FDM group was not significantly different from that in the PBS group ($P > 0.05$), and the number of TH$^+$ cells increased in the 1, 5, and 10 ng groups. The 1 ng group had the highest number of TH$^+$ cells. Therefore, the highest expression of TH$^+$ was achieved through treatment with a low concentration of VEGF165 (Fig. 3B).

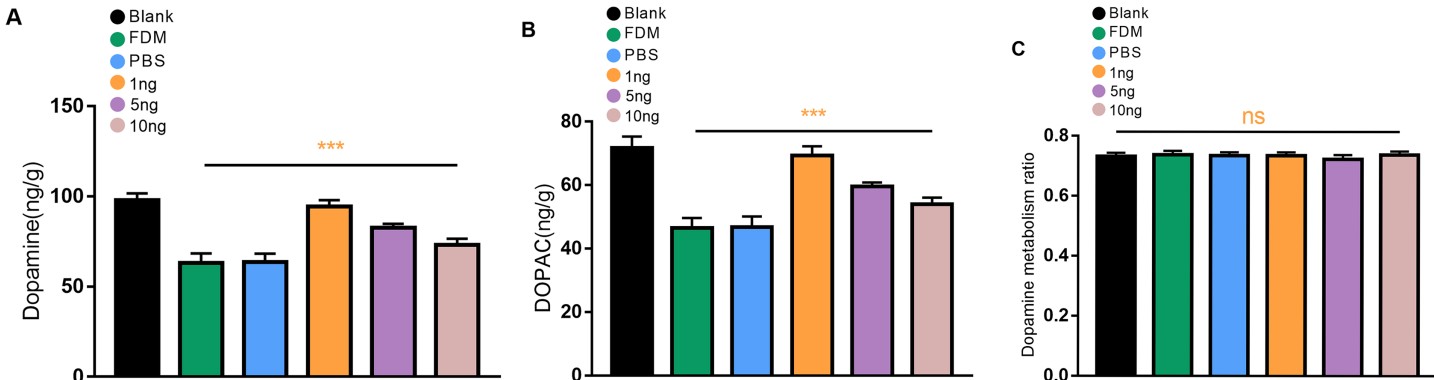

**Figure 2** **Changes in DA (A), DOPAC (B), and dopamine metabolism ratio (C) in the retina after 14 days of intravitreal injection of VEGF165 in FDM guinea pigs.** (A and B) Showed that the level of DA and DOPAC in the retina of FDM guinea pigs were significantly reduced compared with the blank group. After three different doses of VEGF165 were injected into the vitreous of guinea pigs, the level of DA and DOPAC in the retina of FDM guinea pigs were increased, and DA and DOPAC were the most obvious in the 1 ng group ($n = 6$ per group). (C) Means that there was no significant change in the dopamine metabolic rate of each group, and there was no significant significance after comparison of the data of each group ($P > 0.05$). Graphs were depicted as mean ± standard deviation. [***] $P < 0.001$. The data were analyzed using one-way ANOVA followed by a LSD-t test.

## TH/CD31 protein expression level

Western blot analysis was performed to assess retinal TH protein and CD31 protein expression levels after treatment with three groups of VEGF165 at different concentrations. Band of approximately 55 and 130 kDa, corresponding to the molecular weights of TH and CD31, were detected in the retinal protein extracts. The results of Image J gray value analysis showed that TH protein expression was significantly reduced in the FDM group compared with the blank group. DA secretion was reduced in FDM.

The protein detection results of VEGF165 at different concentrations showed that TH expression increased in the three groups, consistent with the above experimental results, and demonstrated the inhibitory effect of VEGF165 on FDM. The highest efficacy was observed in the 1 ng group, and VEGF165 had the most significant effect at low concentrations (Figs. 4A and 4C).

CD31, as a specific marker on vascular endothelial cells, was used in detecting the promoting effect of the intravitreal injection of VEGF165 on neovascularization in the retina. Densitometric analysis did not reveal significant difference in CD31 protein expression level among the three experimental groups (blank, FDM, and PBS). By contrast, in the three groups with different concentrations of VEGF165, the gray value analysis results showed that the expression of CD31 protein increased. The 10 ng group had the highest expression level, whereas and 1 ng group had the lowest. The results indicated that the expression of CD31 protein is positively correlated with the amount of VEGF165 (Figs. 4B and 4D).

## HE staining result of the retina

The results of HE staining of retinal sections showed that the retinas of the guinea pigs in the blank, FDM, and PBS group were neatly arranged, and no or few vascular endothelial cells were distributed in the inner boundary membrane (Fig. 5A). The results of HE

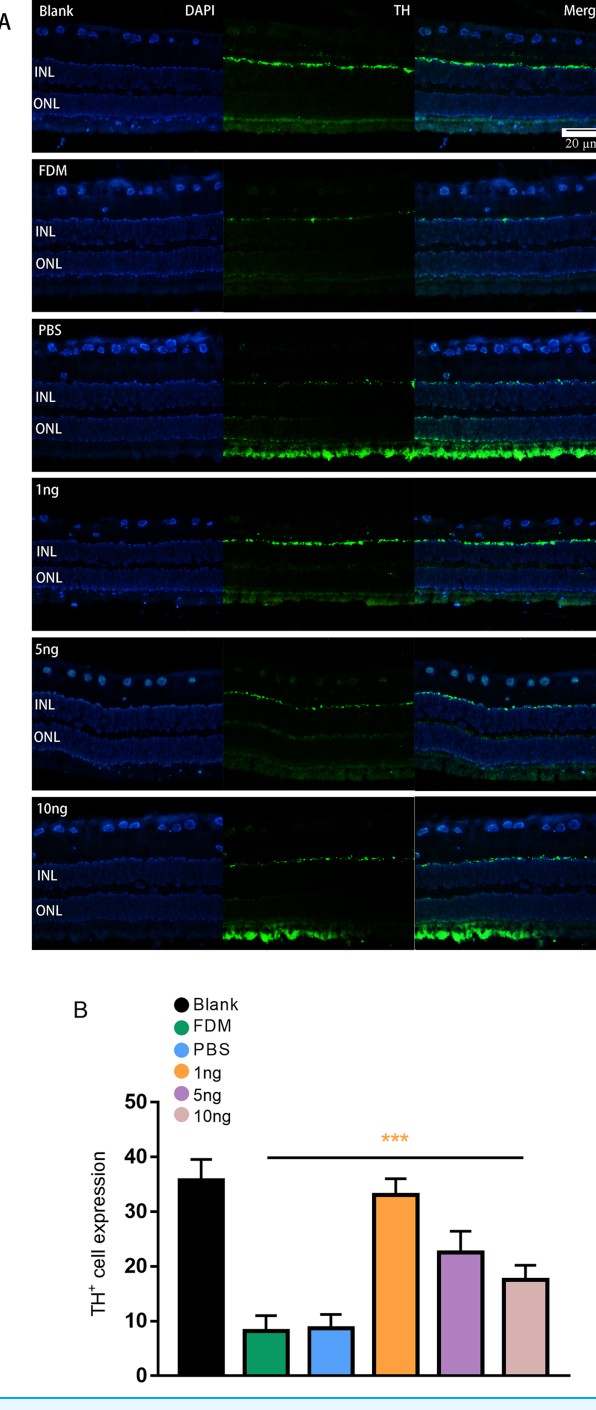

**Figure 3 Significant changes in TH⁺ cells in the inner layer of the retina are observed under fluorescence microscopy.** (A) Represented the changes in the distribution and number of TH⁺ cells under immunofluorescence microscopy in each group. The leftmost row was the image taken under the DAPI channel (blue), the middle row was the image taken under the green fluorescence channel (green), and the rightmost row was the merged picture of the image taken under the two channels. (B) Quantitative analysis showed that there were significant changes of TH⁺ cells in the retina of guinea pigs in each group ($n = 6$ per group). Scale bar: 20 μm. Graphs were presented as mean with ±standard deviation. The data were analyzed using LSD-t test. ***$P < 0.001$.

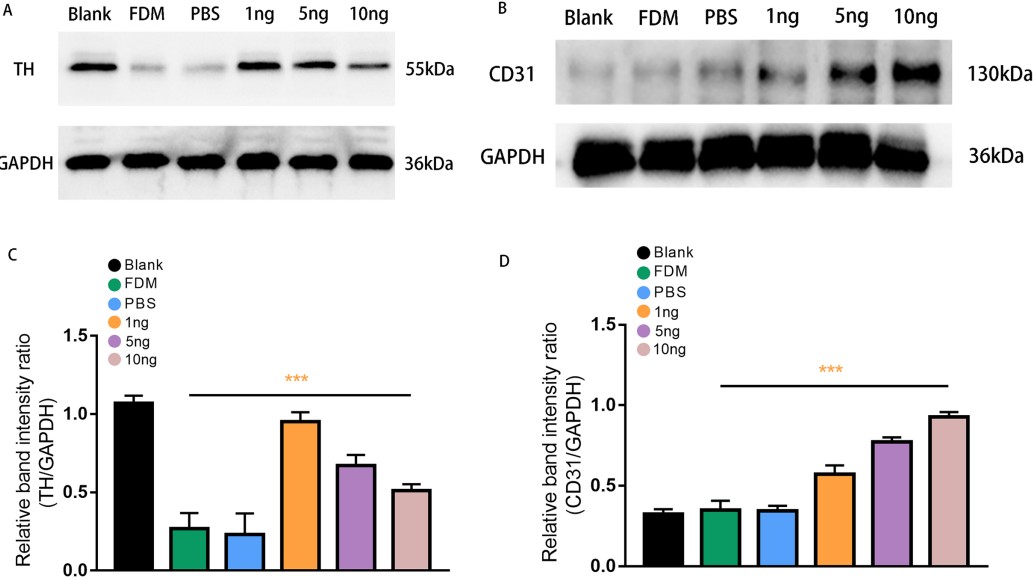

**Figure 4 Expression of TH (A) and CD31 (B) proteins in the retinas of guinea pigs in each group.** (A) Represented TH antibody staining on a Western blotting of retinal protein extracts in different groups. (B) Represented CD31 antibody staining on a Western blot of retinal protein extracts in various groups. (C) TH expression, normalized as a ratio to GAPDH levels, shows the three groups with different doses of VEGF165 were significantly different from the FDM group ($n = 3$ per group). (D) Showed that there was no significant difference between the blank, FDM and PBS groups, while VEGF165 increased CD31 protein expression with increasing dose. Graphs were presented as mean with ±standard deviation. The data were analyzed using LSD-t test. ***$P < 0.001$.   

staining in the three groups with different concentrations of VEGF165 showed increase in number of vascular endothelial cell nuclei in the inner boundary membrane and the number of vascular endothelial cell nuclei increased with the VEGF165 dose. Compared with the blank group, the FDM and PBS groups showed no significant difference in the number of vascular endothelial cell nuclei in the retina ($P > 0.05$), and the number of vascular endothelial cell nuclei in the retina in the 1, 5, and 10 ng groups increased significantly compared with that in the blank group. The differences were statistically significant (all $P < 0.05$). The number of vascular endothelial nuclei in the 10 ng group was the highest, and the number of vascular endothelial nuclei in the 5 ng group was higher than that in the 1 ng group (Fig. 5B).

## DISCUSSION

The incidence of myopia has increased annually in recent years and has become a global public health problem. It has an unclear pathogenesis (*Sarfare, Yang & Nickla, 2020*; *Wu et al., 2020*) and is considered a multifactorial and multisystem complex disease. The exact causes of the myopia epidemic have not been explored, although multiple studies on the causes of myopia have concluded that genetic and environmental factors play an important role in the development of myopia (*Saw et al., 2001*). Environmental factors play an important role in the development and progression of myopia, such as less time outdoors and increase in close work (*Pan, Qian & Saw, 2017*; *Saw et al., 2002*). This role can be

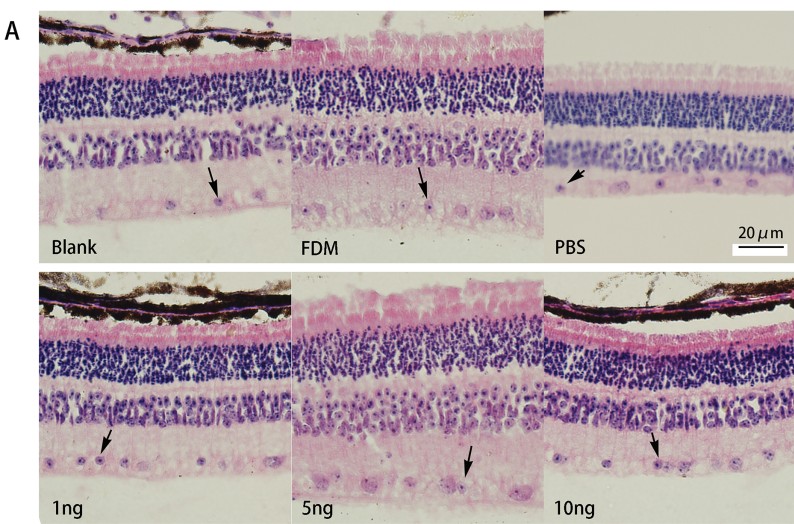

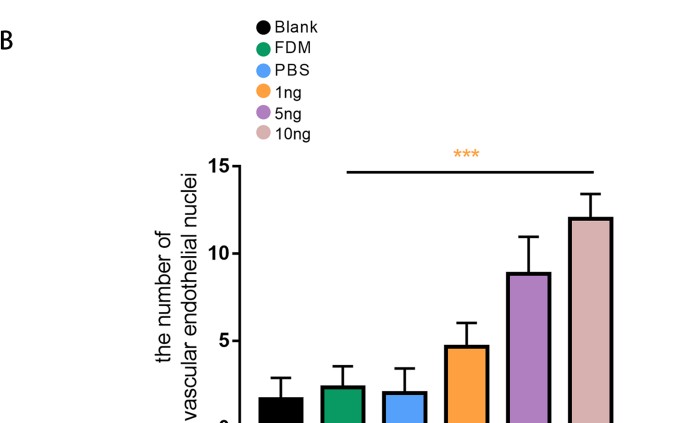

**Figure 5 Changes in the number of nuclei of vascular endothelial cells in the retina of guinea pigs.** (A) Represented the distribution of vascular endothelial cells in the retina after each group of HE staining (*n* = 6, per group), in which vascular endothelial cells were mainly distributed in the inner boundary membrane, the nucleus were blue stained, and the cell bodies were red stained. The black arrows in the figure indicate vascular endothelial cells. (B) Quantitative analysis showed that there was no significant comparison between the blank group, FDM group and PBS group. Compared with FDM group, VEGF165 in all three groups was statistically significant, and 10 ng was the most obvious. Graphs were presented as mean with ±standard deviation. The data were analyzed using LSD-t test. ***$P < 0.001$.

demonstrated through animal models on the regulation of eye growth by visual feedback (*Smith, Hung & Arumugam, 2014*). The successful establishment of the animal models of myopia will allow us to further understand the mechanism of myopia. By establishing an animal model of FDM, DA in the retina was found to be closely related to the occurrence and development of myopia (*Zhou et al., 2017*), the content of DA and DA metabolites decreased in the retina of FDM guinea pigs, and the formation of FDM can be affected by DA content in the retina (*Luo et al., 2017*). DA is a catecholamine neurotransmitter and mainly secreted and released by amacrine cells in the inner layer of the retina (*Sankaran et al., 2018*). Its synthesis originates from tyrosine. After tyrosine is taken up by DA
neurons, it is first converted to levodopa (levodopa–DOPA) under the action of TH and then catalyzed by dopa decarboxylase to form DA. Synthetic DA is decomposed into DOPAC and homovanillic acid (HVA) under the action of monoamine oxidase. TH is a rate-limiting enzyme during DA synthesis, and DOPAC is an important marker reflecting the amount of DA released. This experiment verifies that the content of DA and DOPAC in the retinas of FDM guinea pigs was significantly lower than that in the blank group, and whether the myopia model was successfully constructed was determined by measuring the refraction and AL of the FDM and blank groups. Moreover, the results show that DA participated in the occurrence and development of myopia.

Current myopia treatment methods, such as applying spectacles and contact lenses and performing orthokeratology, can partially control but cannot completely inhibit the progression of myopia. Drug treatment for myopia mainly targets the functions of the retina and sclera signaling molecules or is expected to resolve problems in myopia progression. To date, drugs used to control myopia include low-concentration atropine and parencipine, which have some side effects. Atropine can cause photophobia, myopia blur, loss of adjustment ability, and pupil hyperdilation (*Gwiazda, 2009*). Parenzepine can cause mild-to-moderate side effects, such as cycloplegia, photophobia, mydriasis, conjunctival allergy, and regulatory dysfunction (*Arumugam & McBrien, 2012*). Given that current treatments for myopia are extremely limited, researchers need to find more and better methods to inhibit or prevent the occurrence and progression of myopia. VEGF is a typical neurovascular signal molecule that regulates the vascular and nervous systems, and VEGF exerts neuroprotective effects on DA neurons in the preclinical models of Parkinson's disease. Moreover, and VEGF exerts its neurorescue effect on DA neurons treated with 6-hydroxydopamine *in vitro* and *in vivo* through direct and indirect vascular and neuronal mechanisms (*Yasuda et al., 2007*; *Yasuhara et al., 2005a*, *2005b*). Therefore, we infer that VEGF has neurotrophic and neuroprotective effects on amacrine cells of retina that release DA. In this experiment, changes in DA level in the retina were observed after the intravitreal injection of VEGF165 in the FDM guinea pigs, and the effect of VEGF on DA level in the retina and whether it would inhibit myopia were discussed.

In this study, different doses of VEGF165 were injected intravitreal to verify changes in DA and DOPAC levels in the retina and their effects on FDM. Three doses were used in the experiment (1, 5, and 10 ng). The experimental results showed that compared with the FDM group, the 1, 5, and 10 ng groups showed decrease in myopia in the covered eye, decrease in the elongation of the eye axis, increased DA and DOPAC levels in the retina, increased TH positive expression, and increased TH protein expression, but the ratio of DOPAC to DA remained unchanged. Given that the metabolic rate of DA in the retinas of guinea pigs did not change in each group, increase in DA may be due to increased DA synthesis rather than to a decrease in metabolism. The DA and DOPAC levels, TH-positive cell number, and protein expression in the 1 ng group were higher than those in the 5 and 10 ng group, indicating that VEGF165 had the best inhibitory effect on myopia at low doses. These results are consistent with those of a previous experiment (*Meng et al., 2020*) which explored the effects of VEGF on DA neurons in a rat model with Parkinson's disease after injecting 1, 10, and 100 ng/ml VEGF into the tail vein and after intracranial

injection 1 ng/ml VEGF. The group injected with 1 ng of VEGF showed the best effect on DA neurons in both modes of VEGF injection. Given that *Meng et al. (2020)* studied the neuroprotective effect of VEGF on DA nerves in the brain, we speculate that VEGF exerts a neuroprotective effect on DA level in the retina. In the present experiment, we injected VEGF (1, 5, or 10 ng) into the vitreous of the right eye of each guinea pig to observe the effects of these doses on DA level in the retina. The results showed that VEGF promoted not only DA nerves in the brain but also DA nerves in the retina, and 1 ng of VEGF showed the best effects. The effect of VEGF165 on DA neurons was dose independent, and the highest effects were obtained at low doses. Higher doses can cause excessive angiogenesis and even edema. In the HE staining experiments of the retinal pathological sections of each group, the number of vascular endothelial cell nuclei in the inner boundary of the retina of the 1, 5, and 10 ng groups increased to varying degrees, and the number of vascular endothelial nuclei in the retinas of the 10 ng group was significantly higher than that in the 1 ng group. VEGF is an angiogenic peptide that acts on endothelial cells and promotes the growth of blood vessels (*Neufeld et al., 1999*). CD31 plays an important role in angiogenesis and is expressed in developing and mature vascular endothelial cells, and CD31 is considered a marker for vascular endothelial cells (*Feng, Chen & Zhang, 2016*). Its expression in blood vessels is an indicator for measuring the effectiveness of VEGF (*More et al., 2022*). The Western blot experiment results showed that the protein expression of CD31 responded to changes in vascular endothelial cells in the retinas after different doses of VEGF165 were injected, consistent with the results of HE staining. The expression levels of the CD31 protein in the 1, 5, and 10 ng groups increased to varying degrees, and the level in the 1 ng group was lower than the levels in the 5 and 10 ng groups. The levels in the retinas of the 10 ng group were the largest. The analysis of experimental results can be seen VEGF165 at low doses promoted DA neurons in the retinas, whereas the expression of VEGF165 at higher doses had no obvious effect on DA neurons. On the basis of relevant literature, we infer that the expression of high-dose VEGF can increase the permeability of capillaries and even damage tight junctions between endothelial cells, causing tissue edema. Or VEGF may increase vascular permeability by triggering extracellular $Ca^{2+}$ influx and activating prostaglandin and platelet-activating factors (*Yasuhara et al., 2005b*) and thereby reducing its neurotrophic effect on DA neurons. In rats with oxygen-induced retinopathy, the excessive expression of VEGF damaged the neural structure of the retina and reduced the content of DA in the retina (*Zhang et al., 2013*). Therefore, VEGF dose for intraocular injection is critical.

We administered VEGF into guinea pigs through intravitreal injection. Through multiple experimental methods, we confirmed that VEGF has neuroprotective and nutritional effects on DA neurons in the retina, and DA level in the retina significantly increased. However, owing to limited time and constraints, we were unable to explore the specific mechanism that produces this effect. By reviewing relevant literature, we had the following inferences: First, studies have demonstrated that VEGF directly exerts a neuroprotective effect by binding to receptors flt-1 (VEGF-R1) and flk-1 (KDR/VEGF-R2) and VEGF-R2 can transduce VEGF-induced effects, such as cell proliferation, chemotaxis, changes in protein expression, and anti-apoptotic activity. These effects may rely on the

activation of phosphatidylinositol 3-kinase (PI3-K)/Akt (serine/threonine protein kinase) and mitogen-activated protein kinase (MEK/ERK) signaling pathways to cause anti-apoptotic effects and neuroprotection (*Rosenstein et al., 2003*). Moreover, VEGF promotes neuronal survival by modulating potassium currents, but this effect relies on signaling *via* VEGFR-1. Exogenous increase in VEGF level under hypoxic and glucose-deprived conditions can regulate the level of voltage-gated potassium channel Kv1.2 and promote the tyrosine phosphorylation of this ion channel, promoting cell survival (*Qiu, Zhang & Sun, 2003*). Sema3A can act as a migration inhibitor and apoptotic factor in neural progenitor cell lines, and VEGF165 plays an anti-apoptosis effect by antagonizing the binding of Sema3A to neuropilin-1 (NRP-1) receptors (*Bagnard et al., 2001*). Thus, the third VEGF receptor, NRP-1, plays an important role as a specific receptor for the VEGF165 subtype and a co-receptor for VEGFR-2 (*Soker et al., 1998*) and not only exerts neuroprotective effects in synergy with VEGFR-2 receptors but also exerts neuroprotective effects by competing with Sema3A to bind to the NRP-1 receptors of VEGF165. In addition, VEGF has an indirect neurotrophic effect, not only promoting the formation of blood vessels but also promoting the multiplication of astrocytes. VEGF induces angiogenesis, and vascular endothelial cells secrete neurotrophic factors, such as brain-derived neurotrophic factor (BDNF), creating a BDNF gradient that favors neuronal migration. Moreover, activated endothelial cells can provide humoral direction and maintain physical scaffolds for neuronal movement (*Louissaint et al., 2002*). Astrocytes produce a large number of neurotrophic factors, such as glial-cell-derived neurotrophic factor (GDNF) to support nerve growth and survival, especially DA neurons (*Silverman et al., 1999*). VEGF plays a role in retinal development, and the application of VEGF *in vitro* increases the number of photoreceptors and amacrine cells (*Yourey et al., 2000*), and thereby promotes the secretion of DA in the retina and inhibiting myopia. However, the specific mechanism needs experimental evidence. In the next stage, we will systematically discuss the mechanism and detailed pathway of the specific neuroprotective effect of VEGF to lay a solid theoretical foundation for the early application of VEGF in myopia prevention and control.

## CONCLUSION

The intravitreal injection of VEGF165 can protect dopaminergic neurons in the retina and increase the content of DA and DOPAC in the retinas of FDM guinea pigs, thereby inhibiting the development of refraction and eye AL and the formation of myopia. In the FDM guinea pig models, low doses of VEGF165 exerted significant neuroprotective effects on DA neurons in the retina through angiogenesis and direct action on DA neurons. Therefore, VEGF165 can be used in the prevention and control of myopia and provide experimental basis for the prevention and treatment of myopia.

## ACKNOWLEDGEMENTS

We gratefully thank the contributions from Medical Research Center of Binzhou Medical University Hospital for their assistance of experimental equipment and technology.

### Funding
The authors received no funding for this work.

### Competing Interests
The authors declare that they have no competing interests.

### Author Contributions
- Ruiting Sun performed the experiments, authored or reviewed drafts of the article, and approved the final draft.
- Qingsheng Peng performed the experiments, authored or reviewed drafts of the article, and approved the final draft.
- Fengyi Zhang analyzed the data, authored or reviewed drafts of the article, and approved the final draft.
- Honglian Gao analyzed the data, prepared figures and/or tables, and approved the final draft.
- Tong Li analyzed the data, prepared figures and/or tables, and approved the final draft.
- Lei Wang analyzed the data, authored or reviewed drafts of the article, and approved the final draft.
- Lei Zhang conceived and designed the experiments, authored or reviewed drafts of the article, and approved the final draft.

### Animal Ethics
The following information was supplied relating to ethical approvals (*i.e.*, approving body and any reference numbers):

The study was approved by the ethics committee of the Affiliated Hospital of Binzhou Medical University (approval number: 20220128-78)

### Data Availability
The raw measurements are available in the Supplemental File.

### Supplemental Information
Supplemental information for this article can be found online at http://dx.doi.org/10.7717/peerj.16255#supplemental-information.

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
