# Peer review of "Effect of vascular endothelial growth factor 165 on dopamine level in the retinas of guinea pigs with form-deprivation myopia"

_PeerJ, doi:10.7717/peerj.16255_

## Round 0.1 · original submission · Major Revisions

Please address concerns of all reviewrs and amend your manuscript accordingly.

Reviewer 1 ·

Basic reporting

1. Language: The paper is written in a professional English language. However, there are still several ambiguities or typos. To name a few, in line 23, comma between “myopia” and “Vascular” is inappropriate. In line 112, the notation “1, 5, and 10 ng/2 μL” is confusing. The sentence in line 136 is grammatically wrong. In line 322, “but DOPAC or DA remained unchanged” seems to be wrong – I guess it should be “ratio of DOPAC to DA remained unchanged”. The complete manuscript should be thoroughly proofread before publishing by authors. Such kinds of ambiguities or typos are not allowed.
2. Structure: The paper is clearly structured and conforms to PeerJ standards.
3. Figure: Legends are missed in figure 3A, 4A and 4B, 5A. In 3A and 5A, it’s better to denote what does each color mean in images. In 4A and 4B, labels for all bands and their kDa number are required.
4. Raw Data: Raw data are provided as required.
5. Abstract and Introduction: Abstract and introduction are well written with all necessary background provided. However, the authors mentioned “Parkinson’s disease” in abstract, which has nothing to do with the whole paper and is very confusing to reader.
6. Discussion and Reference: All literature is well referenced. However, discussion part focuses too much on literature review rather than results of this paper. More discussion of results should be added, and literature review should be shorten - Only the most critical literature review can be retained.
7. Acknowledgement: It’s inappropriate to not state acknowledgement is not applicable, except the author covers all the cost of this work. Credits should be given to funding, people providing useful suggestions and help.

Experimental design

1. This work provides a good confirmation of the hypothesis that VEGF165 can help prevent myopia through control DA level. It would be better if authors can provide more insights about the mechanism of action of VEGF165 on DA in the retina as they stated that such mechanism is unclear. This will add more novelty and impact of this paper.
2. Although VEGF165 promotes the growth of vascular endothelial cell, higher dose of VEGF165 (such as 5 ng and 10 ng), and thus higher expression of vascular endothelial cell, fails to produce as much DA as lower dose of VEGF165 (such as 1 ng) and prevent myopia. It would be better if more experiments/discussion could focus on exploring underlying mechanism.
3. All the instruments with detailed model should be provided in methods part. For example, line 168 only states rpm without size of centrifugal machine, which makes this work hard to be reproduced if centrifuge is important here.

Validity of the findings

1. All data have been provided and analyzed. All the analysis in results part is robust, statistically sound and controlled according to figures provided.
2. Conclusions are well stated.

Additional comments

This paper studies the influence of vascular endothelial growth factor 165 (VEGF165) on retina dopamine (DA) level and the consequent effect on myopia prevention and control. An optimal dose of VEGF165 of 1 ng is reported to give the highest DA level and best inhibitory effect on myopia. Based on all the comments above, I would suggest a major revision. This manuscript may be considered for acceptance once all the aforementioned comments have been thoroughly addressed.

Reviewer 2 ·

Basic reporting

In this paper, the writing is generally good, with clear and easy-to-understand ideas. However, there are still a few small issues that could be improved:
Line 73, before “but also” should be a comma instead of full stop
Line 75 -76, since the author mentioned that the subtype of VEGF, VEGF-A is often referred to as VEGF, the author is better to clarify that in this paper VEGF is actual VEGF or VEGF-A
Line 315-325, too wordy, the author mentioned one thing increased and doesn’t have to mention the other thing decrease, too wordy.
Figure 2 Need to be more clear with Figure A and B, which is DA and which DOPAC, the author can change the title like “Changes in DA (Fig.2A), DOPAC(Fig.2B), and dopamine metabolism ratio (Fig.2C) in ….”
Figure 4, in A and B, need to label each gel band, or make clear note like “the band from left to right: Blank, FDM, PBS, 10ngVEGF…..” Also need to clarify in figure A and B that top gel are TH or CD31, and bottom gel are GAPDH

Experimental design

Figure 4A, and 4B, the author mentioned loaded same amount protein homogenate in each lanes (20ug), however, from the supporting information, WB folder, and subfolder of CD31, the original whole gel for figure4B, there are multiple bands on each lane, especially a strong lower MW band showed in each lane, if the author load same amount protein, this low MW “impurity” should have similar intensity. Based on this whole gel, looks like the author loaded different amount of proteins for different lanes, could you explain this?
Figure 5, does the author calculate the number of cells based on the area? If so, the author need to put scale bar in the figure to indicate they are same scale, and also clarify in the method section.

Validity of the findings

Line 329-331, when the author compare the result with reference (Meng et al. 2020), the author should briefly mention how this work is done, maybe give more discussion and explainaiton, more reference in this part. The reference by Meng at al used 1-100 ng/mL dose, however, this paper used 1-10 ng/uL, which is 1/10 amount as for the reference paper, so it is not proper to use the word “highest dose” or “lowest dose”
Line 337-338, the author mentioned that expression of CD31 is an indicator for measuring the effectiveness of VEGF, the reference Newman 1997 doesn’t seem to talk about this? the author should cite a proper reference and also brief discuss the what kind of effectiveness of VEGF is related to CD31expression.
Line 345-350, this part is not explained very well, doesn’t make much sense. The author first clarified the higher dose of VEGF increase CD31 cells, which indicated more vascular endothelial cell, but this higher dose (10ng) actually have similar amount of DA and DOPAC, compared to Blank group (Figure 2A and 2B). The lower dose (1ng and 5ng) VEGF Still increased CD31 expression but with a lower level, however, while the actually amount DA and DOPAC is lower compared with blank group (Figure 2A and 2B). The author needs to focus more one this part discussion, instead of just repeatedly stating the experiment result.
Line 351 -361, the author seems just give some examples about VEGF receptors. It would be more meaningful to have a deep discussion about how VEGF interact with the receptor and trigger downstream regulatory pathways, and thus play roles in the proliferation and survival of vascular endothelial cells

Additional comments

Overall, the paper presents clear and easy-to-understand ideas. However, a significant portion of the discussion section is devoted to background information, and there is repetitive emphasis on the experimental results. It would greatly enhance the paper if the author could delve deeper into how previous studies support this work or how this work complements existing research. Additionally, exploring the intricate relationships between VEGF, vascular endothelial cells, and DA (dopamine) would provide valuable insights for the readers

Reviewer 3 ·

Basic reporting

Ruiting Sun and the colleagues reported suppressive effects of VEGF165 intravitreous injections against form-deprivation myopia of guinea pigs in a dose-independent manner. Their findings are interesting; however, several points need to be improved with a revision.

Major points:
1. The most important vascular tissue in the eye for myopia progression is the choroid rather than the inner retina. The authors should evaluate the morphological change of the choroid after VEGF165 injections.
2. The vascular endothelial cells cannot be identified in the HE staining (Figure 5). They should show immunostaining with endothelial markers such as CD31.

Minor points:
1. All results should be presented as the following order: blank, FDM, PBS, 1ng, 5ng, and 10ng. This is because 1ng is the lowest dose next to the PBS group.
2. In methods, describe the details of quantification for TH+ cells (Fig. 3B).

Experimental design

See above.

Validity of the findings

See above.

Additional comments

None.

---

## Round 0.2 · accepted · Accept

All concerns of the reviewers were addressed and the manuscript was amended accordingly. Therefore, the revised version is acceptable now.

Reviewer 1 ·

Basic reporting

no comment

Experimental design

no comment

Validity of the findings

no comment

Additional comments

The author addressed all the comments I made earlier. The revised version is much improved. I recommend acceptance.

Reviewer 2 ·

Basic reporting

The writing and figure annotation has been improved.

Experimental design

The author has explained the reason for the inconsistency of Figure 4B, which seems to be a technique issue. The author or the group should improve this in the future, and provide the best representative figures.

Validity of the findings

The author has improved the discussion and provided more insights about the mechanism of action of VEGF165 on DA in the retina.

Additional comments

Overall the manuscript has been significantly improved and can be accepted as it is.